# OUTLIERS MEMORIZED LAST:
# TRENDS IN MEMORIZATION OF DIFFUSION MODELS BASED ON TRAINING DISTRIBUTION AND EPOCH

## ABSTRACT

Memorization and replication of training data in diffusion models like Stable Diffusion is a poorly understood phenomenon with a number of privacy and legal issues tied to it. This paper analyzes how the location of a data point in the training dataset's distribution affects its likelihood of memorization over training epochs. Importantly, it finds that memorization of 'outliers' is less likely early in the training process until eventually matching with the rest of the dataset. It then suggests applications utilizing this difference in memorization rate, including hyperparameter tuning and anomaly detection. It then suggests research that could be done from this conclusion to further improve memorization understanding.

## 1 INTRODUCTION

Diffusion models are a class of generative neural networks that utilize iterative denoising on a sample from a trained distribution in order to produce an image.Ho and et al. (2020) They are a rapidly emerging model in computer vision, with Open Diffusion, Imagen, and DALLE-2 being notable examples. However, these large-scale models have a well-documented issue: memorizing and reproducing individual training examples.Carlini and et al. (2023) This violates all guarantees of privacy and causes a number of issues related to copywriting and "digital forgery".Somepalli and et al. (2022) In this paper, I seek to better understand this phenomenon as it relates to their training distribution. By training a two-feature toy diffusion model hundreds of times with a variety of different distributions, I was able to measure memorization over epochs at the tail vs. the head of the training distribution. I found clear differences in the rates of memorization between the two. Specifically, training examples at the tail/outlier of the training distribution were memorized at a slower rate early in the training process.

## 2 RELATED WORK

In terms of memorization being researched with respect to outliers, the most relevant work is that of Vitaly Feldman.Feldman (2021) This paper, while making interesting connections between learning and memorization of atypical examples, does not explore the relationship with respect to the training process. There does exist research on the distinction between memorization of near duplicates and the rest of the distribution, with findings showing that they are on average memorized at a higher rate.Carlini and et al. (2023) It is also important to note a clear distinction has been found between memorization and overfitting. Memorization has been identified as a separate phenomenon altogether.Neyshabur et al. (2017) However, there do not seem to be any papers that explore the relationship between memorization and the training process itself over epochs. This paper aims to begin filling that gap.

## 3 DEFINITIONS

In order to better understand this phenomenon, I had to make a number of assumpitons in the form of definitions of the metric of memorization and our seperation of the training distribution.They are listed below.

## 3.1 MEMORIZATION

It is increadibly difficult to perfectly encapsulate memorization using a metric. While previous metrics have used black box extraction (Carlini and et al. (2023)), memorization is defined here using a more intuitive reasoning. It can be argued that examples that closely fit a training example while being far from any test example is memorized, since rather than being generalized to the distribution it is replicating the training example. Therefore, we define memorization as follows: **A generated example is memorized if it is much closer(as defined by a 2:1 ratio) to its closest training example than its closest test example.**

## 3.2 INLIERS

Inliers are defined as the majority of the training distribution. In the case of the toy diffusion model, this is the entire original training set without the injected outliers or near duplicates.

## 3.3 OUTLIERS

Outliers aim to represent the tail of the distribution. Here, they are defined as training examples injected into the training set that are far from the original distribution. This is done for the toy dataset by generating a random cluster of points that are 2 standard deviations from the mean of the original distribution.

## 3.4 NEAR DUPLICATES

Duplicates are defined as training examples that are very close or at the same position as each other. This is done by sampling random samples in the original distribution and adding points equal to those samples.

## 4 EXPERIMENT

My aim was to determine if the following was true: **Diffusion models memorize more duplicates(as placed into the dataset) at an earlier training epoch while memorizing outliers (as placed into the dataset) at later epochs.** To set up the experiment, I trained a diffusion model on a toy dataset with two features. The initial skeleton for the model was taken from(Tanelp (2023)). This public github repo features a very simple toy model that was easy to train and iterate on. This dataset was a mixture of two moons with a small amount of noise. By adjusting hyperparameters, I was able to induce a high rate of memorization in the initial model. It is currently not directly linked due to anonymity rules. I then trained models with the same hyperparameters, except I injected a set amount of outliers and near duplicates as defined above into the training set. At the beginning of each training epoch, I also had the model generate a set of examples of the same size as the training set. After training, these generated sets at each epoch were tested for memorization. This was done using FAISS to determine, for each generated point, the samples they were closest to in both the test and training set. If the closest training sample was significantly closer than the closest test sample, it was considered memorized, as per the definition above. At each epoch of training, I generated examples equal to the training set size. Each generated example was then compared to all training and test examples to find the closest to each at the given epoch. These closest points where then used to identify a memorized point as defined in the previous section. The memorized points at each epoch were then classified by whether they were an injected near duplicate, an injected outlier, or an already existing inlier. These statistics were then used to plot the graphs depicted below.

## 4.1 PERCENTAGE OF MEMORIZATION OVER EPOCHS

These graphs show the percentage of memorized points as a fraction of the total size of the generated set, mapped over each epoch, with varied amounts of injected outliers,injected near duplicates, and epochs. This expirement was tested with these same paramaters 10 times each, with random training distributions as defined above each time. Similiar results with the same general trends were seen.

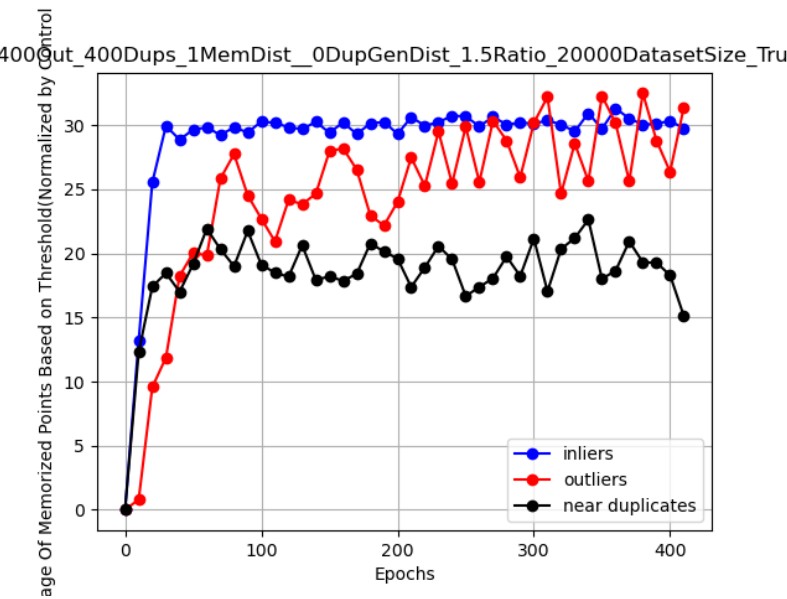

Figure 1: 400 Inliers/400 Outliers Injected, 400 Epochs, 20000 Dataset Size, 1.5 Ratio Between Test and Train Data For Determining Memorization

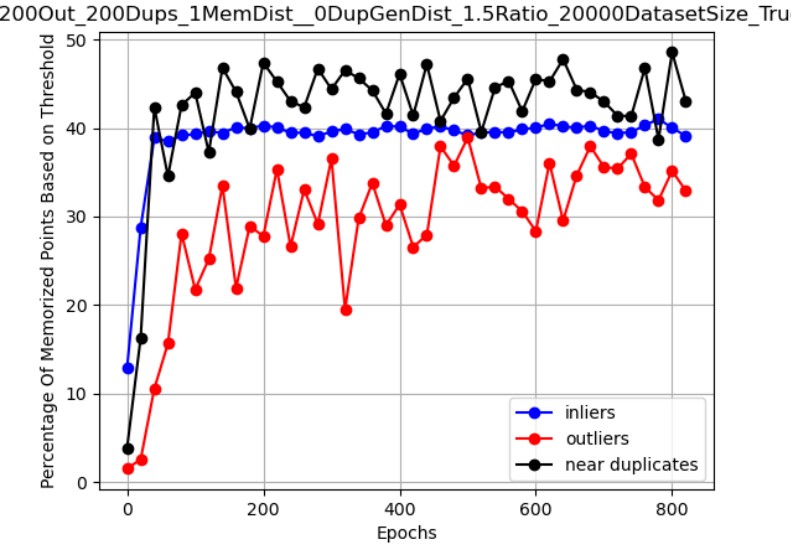

Figure 2: 200 Inliers/200 Outliers Injected, 800 Epochs, 20000 Dataset Size, 1.5 Ratio Between Test and Train Data For Determining Memorization

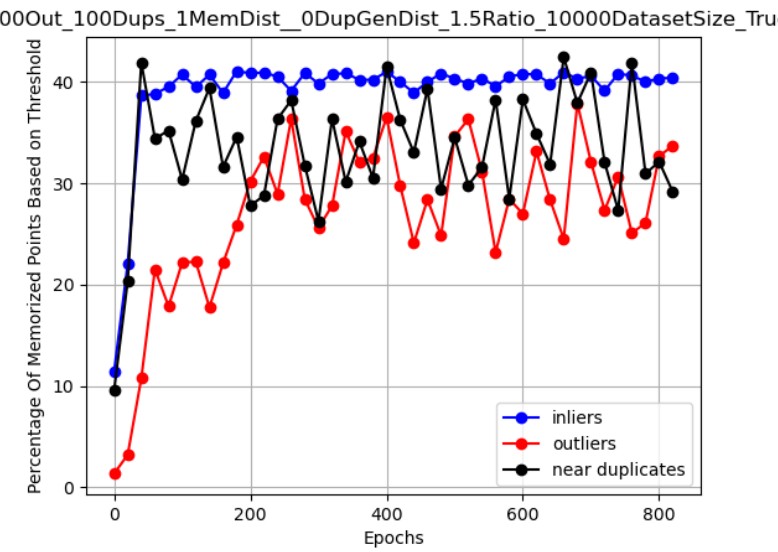

Figure 3: 100 Inliers/100 Outliers Injected, 800 Epochs, 10000 Dataset Size, 1.5 Ratio Between Test and Train Data For Determining Memorization

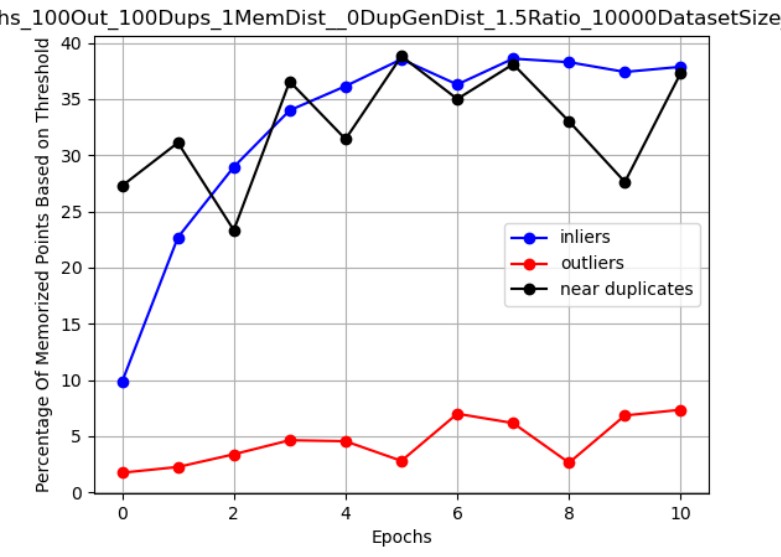

Figure 4: 100 Inliers/100 Outliers Injected, 10 Epochs, 10000 Dataset Size, 1.5 Ratio Between Test and Train Data For Determining Memorization

## 5 ANALYSIS AND DISCUSSION

Through all the data, a common trend can be seen: memorization of injected outliers happens later than the rest of the distribution. This difference is also universal with changes to the thresholds, injection size, and training data size. This is a property of the training process itself, and not the model. This trend is an important step forward in our understanding of memorization in diffusion models, while also opening up potential useful applications in anomaly detection and tuning.

### 5.1 APPLICATIONS

Outliers being memorized later help in identifying images with uniform outlier features: images with watermarks, for example, tend to be a large outlier cluster, especially if the distribution is differentiated over a focused subset of features. By identify the group of examples that are memorized later than the rest of the distribution, we would be able to identfiy and parse watermarks through a diffusion model training proccess. Tuning hyperparameters based on the memorization of outliers could also be useful for fine-tuning GANs. Outliers beginning to be memorized could be an indicator for overfitting starting, allowing early stops: if data you know is an outlier start to be memorized early stop.

## 6 LIMITATIONS AND FUTURE WORK

This experiment has yet to be done on anything beyond the toy diffusion dataset and thus requires testing on more varied distributions, as well as datasets with more features. Testing to see if this is a property of the training process or the model itself is highest in importance. Doing so will require a more computationally efficient method of identifying memorization, as doing the closest point comparisons would not scale to large models. Another important step would be finding what the relationship between model generation quality and memorization is, in order to understand how this property relates to model learning.

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
