# OpenReview forum: "Outliers Memorized Last: Trends in Memorization of Diffusion Models Based on Training Distribution and Epoch"
_ICLR.cc/2024/Conference — Submitted to ICLR 2024_

### Official Review · Reviewer_uuB7 · 2023-10-27

**Soundness:** 1 poor
**Presentation:** 1 poor
**Contribution:** 1 poor
**Rating:** 1
**Confidence:** 4

**Summary:**

Engaging in  discussion on memorization within generative models, but set entirely against a
 synthetic toy data backdrop. The absence of explicit definitions for terms like inliers, outliers, and memorization stands out. It is a preliminary study and calls for diving deeper into the empirical analysis.

**Strengths:**

The topic of memorization in diffusion models is interesting and requires in depth study.

**Weaknesses:**

- The claims and definitions are unclear.
- Experiments are carried out only on an arbitrary synthetic dataset

**Questions:**

NA

---

### Official Review · Reviewer_c1Ba · 2023-10-31

**Soundness:** 2 fair
**Presentation:** 2 fair
**Contribution:** 2 fair
**Rating:** 3
**Confidence:** 4

**Summary:**

Diffusion models have attracted significant attention due to their ability to generate high-quality synthetic images. However, recent research discovers that diffusion models are likely to memorize training images, i.e., produce identical training images during the inference phase. In this paper, the authors aim to undermine the relation between memorization and train distribution. They observe that memorization of ’outliers’ is less likely early in the training process until eventually matching with the rest of the dataset.

**Strengths:**

The discussed topic -- memorization of diffusion models is important and interesting.

**Weaknesses:**

- The content of the paper is not substantial. The current version has only four pages.
- Lack of experiments on real data. The experiments consider only synthetic dataset, which makes the observation less convincing.
- SOTA text-to-image diffusion models, e.g., stable diffusion, are not discussed in this paper.

**Questions:**

Please refer to the weaknesses.

---

### Official Review · Reviewer_FcWa · 2023-11-02

**Soundness:** 2 fair
**Presentation:** 1 poor
**Contribution:** 2 fair
**Rating:** 1
**Confidence:** 5

**Summary:**

This paper explores the impact of the position of a data point within a training distribution on memorization rates across epochs in diffusion models. A simplistic 2-feature toy dataset is used to assess the memorization of both injected outliers and inliers, along with near duplicates, throughout the training process.

A key discovery is the slower rate of outlier memorization during the early training stages, which, with time, aligns with the memorization rate of other data points. I think this is an interesting finding which supports a similar result in LLM literature by Tirumala et. al. https://arxiv.org/abs/2205.10770

However, this is only a preliminary study, calling for additional empirical analysis to validate these findings and delve into the underlying causes in practical scenarios. The purported applications also warrant further validation to ensure their efficacy.

**Strengths:**

1. The exploration of the impact of a data point's position on its memorization is an intellectually interesting question, and also has practical implications centered around privacy.
2. The finding that there is slower rate of outlier memorization during the early training stages is interesting.

**Weaknesses:**

1. This paper does not compare with related work.
2. The draft is a very preliminary compilation of results on a toy setup and its quality is below that of a conference paper.
3. Discussion with https://arxiv.org/abs/2205.10770 might be helpful.

I encourage the authors to continue working on the same and present it as a solid and generalizable result across different input complexities.

**Questions:**

N/A

---

### Official Review · Reviewer_ehcz · 2023-11-02

**Soundness:** 1 poor
**Presentation:** 1 poor
**Contribution:** 1 poor
**Rating:** 1
**Confidence:** 5

**Summary:**

The paper investigates memorization in diffusion models, and in particular how in-distribution, out of distribution/outlier and duplicate points are memorized. To do so, the authors train a toy model on a toy dataset with different numbers of outliers and duplicates. They find that in-distribution and duplicate points are memorized earlier, and outliers later during training.

**Strengths:**

Memorization in generative models is a highly relevant topic. The paper approaches an interesting problem in this space, by investigating how the memorization dynamics during training depend on factors such as likelihood under the training distribution and frequency/duplication inside the training dataset, in the context of diffusion models.

**Weaknesses:**

- The results and the methodology in the paper are very preliminary. All experiments in the paper use a single toy diffusion model, as well as a single toy synthetic dataset. While such an approach can be valuable to form initial hypothesis about how models might memorize in more realistic settings, they need to be followed up with an extensive validation on more realistic datasets and realistic and diverse model architectures (or a thorough theoretical analysis). Otherwise it is not clear whether the findings from the toy setting will translate to practical scenarios.
- The paper uses a questionable notion of memorization. A datapoint is considered memorized if it is closer to a training, rather than a test example. This definition of memorization has several issues:
    - It does not consider model behavior or internals in any way. It will make the same memorization judgements for any models, even if they behave very differently. This is not realistic, as memorization should somehow depend on the model.
    - It strongly depends on the sampling procedure of the training and test sets. In particular, whether a point is deemed memorized is a function of how many and how densely test points are sampled, while keeping the training set fixed.
    - There is no formal definition of memorization, and it's not clear what distance metric is used and how the 2:1 ratio in the definition is applied.
- Many details of the experimental setup are not specified, making it difficult to interpret the results:
    - The dataset used in the experiments is not properly described. The text mentions two features as well as two moons with some noise, but does not provide a description or visualizations of samples from the dataset beyond that. Is the dataset a 2D image dataset or does it consist of just two numbers per data point? What sampling parameters are used? How are outlier points generated? Is the dataset based on prior work or was it created by the authors?
    - What hyperparameters are used for the model architecture and training: how many layers, what hidden size, how many parameters in total, how many diffusion timesteps, what are the optimization parameters, etc.?
    - What are the differences between the training runs in Figure 1 - 4? Only the number of inliers, outliers, training epoch and the dataset size, or are other parameters varied as well? Why were those combinations of hyperparameters investigated, i.e. why are they of particular interest?
- Some minor points:
    - The citation format is a bit strange and does not seem to correspond to the style guide.
    - The paper is written in the "I" form. While this is a matter of taste, it is rather unusual.

**Questions:**

- What does it mean to have e.g. 100 inliers in Figures 3 and 4, but 10000 dataset points? According to the definition in 3.2 inliers are the majority of the training points, so it's not clear what the difference is.
- In Section 5 you say that the observed memorization behavior is a property of the training process and not of the model. How can you know that when you only investigate one type of model?

---

### Meta-Review · Area_Chair_aGAH · 2023-12-06

**Metareview:**

This paper explores an important topic of memorization in generative models, focusing on the dynamics of memorization during training in the context of diffusion models. The reviewers recognize the relevance of the problem and appreciate the exploration of factors such as likelihood under the training distribution and frequency/duplication inside the training dataset. A reviewer found value in the finding “slower rate of outlier memorization during early training stages”. There unanimous consensus among reviewers is to reject the paper due to the preliminary nature of the study which lacks sufficient empirical analysis to validate the findings and explore underlying causes in practical scenarios. The authors did not provide a rebuttal and the overall consensus of reviewers remained the same.

**Justification For Why Not Higher Score:**

The paper's findings are preliminary and more work is needed to make it publication-worthy.

**Justification For Why Not Lower Score:**

N/A

---

### Decision · Program_Chairs · 2024-01-16

Reject